# Circulating miR-122-5p, miR-92a-3p, and miR-18a-5p as Potential Biomarkers in Human Liver Transplantation Follow-Up

**DOI:** 10.3390/ijms24043457

**Published:** 2023-02-09

**Authors:** Cristina Morsiani, Salvatore Collura, Federica Sevini, Erika Ciurca, Valentina Rosa Bertuzzo, Claudio Franceschi, Gian Luca Grazi, Matteo Cescon, Miriam Capri

**Affiliations:** 1DIMEC-Department of Medicine and Surgery, University of Bologna, Via S. Giacomo 12, 40126 Bologna, Italy; 2Interdepartmental Center, Alma Mater Research Institute on Global Challenges and Climate Change (Alma Climate), University of Bologna, 40126 Bologna, Italy; 3Hepatobiliary and Transplant Surgery Unit, IRCCS, Universitaria Sant’Orsola Hospital, University of Bologna, 40138 Bologna, Italy; 4Laboratory of Systems Medicine of Healthy Aging, Department of Applied Mathematics, Lobachevsky University, 603022 Nizhny Novgorod, Russia; 5Hepatopancreatobiliary Surgery, IRCCS-Regina Elena National Cancer Institute, 00114 Rome, Italy

**Keywords:** microRNAs, noninvasive biomarkers, liver transplant, follow-up timing

## Abstract

The requirement of blood-circulating sensitive biomarkers for monitoring liver transplant (LT) is currently a necessary step aiming at the reduction of standard invasive protocols, such as liver biopsy. In this respect, the main objective of this study is to assess circulating microRNA (c-miR) changes in recipients’ blood before and after LT and to correlate their blood levels with gold standard biomarkers and with outcomes such as rejection or complications after graft. An miR profile was initially performed; then, the most deregulated miRs were validated by RT-qPCR in 14 recipients pre- and post-LT and compared to a control group of 24 nontransplanted healthy subjects. MiR-122-5p, miR-92a-3p, miR-18a-5p, and miR-30c-5p, identified in the validation phase, were also analyzed considering an additional 19 serum samples collected from LT recipients and focusing on different follow-up (FU) times. The results showed significant, FU-related changes in c-miRs. In particular, miR-122-5p, miR-92a-3p, and miR-18a-5p revealed the same trend after transplantation and an increase in their level was found in patients with complications, independently from FU times. Conversely, the variations in the standard haemato-biochemical parameters for liver function assessment were not significant in the same FU period, confirming the importance of c-miRs as potential noninvasive biomarkers for monitoring patients’ outcomes.

## 1. Introduction

Liver transplant (LT) represents the gold standard intervention for acute and chronic end-stage liver diseases, the only therapeutic option in the case of complete liver failure. LT may improve quality of life and significantly extend life expectancy, according to the age of the patients [1]. On the other hand, the shortage of human donor organs moves toward the increased use of marginal donors, including organs from old or very old donors usually transplanted into younger recipients [2]. There has been remarkable progress in LT outcomes over recent decades, largely in virtue of better organ storage after donor explant, perioperative/postoperative procedures, immunosuppressive treatments and as a result of advances in the comprehension of several liver diseases. Nevertheless, hepatic and systemic complications are still common in the early and long-term period after LT, hence, resulting in significant morbidity and mortality [3]. The short-term complications include graft dysfunction and acute rejection, infections, and systemic damage. On the other hand, long-term complications often arise as a consequence of either the recurrence of primary liver disease or immunosuppressive treatment, counting mainly chronic rejection, renal failure, and recurrent and de novo malignancy. Furthermore, metabolic complications are very common among LT recipients, being the main cause of cardiovascular morbidity and mortality in the long-term follow-up of LT patients [4].

Clinical and haemato-biochemical analyses of patients after transplantation with histopathological analysis of liver biopsies have been the standard method for the detection of liver injuries so far. However, liver biopsy procedures have limitations, including their invasive nature with the possibility of complications and sampling variability [5]. Therefore, noninvasive methods and the use of biomarkers for keeping transplanted patients under observation are broadly considered a more advantageous strategy in comparison with the adoption of graft biopsies [6,7]. In particular, specific enzymes and end products of hepatic metabolism are considered useful markers of organ damage and dysfunction [8]. Laboratory liver function tests (LFTs) are commonly used as clinical tools for the monitoring of patients affected by hepatic dysfunction and transplantation. These tests consist of measuring and assessing various liver function markers, such as serum bilirubin, albumin, alanine aminotransferase, aspartate aminotransferase, ratio of aminotransferases, alkaline phosphatase, gamma-glutamyl transferase, prothrombin time and international normalized ratio, 5′-nucleotidase, ceruloplasmin, and α-fetoprotein. Even if these measurements are still standardly adopted, sometimes they appear to lack specificity and sensitivity [9]. Some studies also highlighted significant age-related changes in the LTFs, but the data displayed a great heterogeneity in the analyzed cohorts [10], thus reinforcing the need of new circulating biomarkers with higher specificity and sensitivity for liver function.

MicroRNAs (miRs) have gained enormous interest in the field of biomarker research. MiRs are a class of small noncoding RNAs, 18–24 nucleotides in length, able to regulate gene transcription, and evidence confirms their involvement in liver aging [11] and the pathogenesis of liver diseases and transplantation [11,12,13,14].

The primary endpoint of this study focused on blood-circulating miR (c-miR) changes in transplanted patients to evaluate their possible role as biomarkers of successful LT and their modification during LT follow-up according to different times (from 4 up to 26 months). The secondary endpoint of the study was to analyze the previously identified miRs of liver aging (i.e., miR-31, miR-141, and miR-200c) as potential biomarkers of liver aging/rejuvenation in the blood of recipients [11].

## 2. Results

### 2.1. Discovery Phase and miR Profiling

The discovery phase and miR profiling were performed to investigate the potential changes in the miR expression of the recipients’ serum in pre- vs. post-LT, comparing the same recipients. The analysis was carried out using three pairs of samples obtained from three recipients (as reported in Table 1, patients highlighted in bold), each of them collected before LT and after 4–8 months of FU. Fourteen miRs were identified according to a fold change ≥ 2 and ≤−2, as reported in Figure 1. The three most up- and downregulated miRs (six miRs, see below) were selected for subsequent validation analysis.

### 2.2. Validation Phase of Selected miRs by RT-qPCR

The validation phase was conducted by means of RT-qPCR on the same serum samples used for the miR profiling. Six miRs were selected for validation (i.e., miR-381-3p, miR-122-5p, miR-518f-3p, miR92a-3p, miR-30c-5p, and miR-18a-5p), since they showed the highest variation before and after the liver transplantation in the discovery phase.

Due to the high Ct values (Ct > 32) of miR-381-3p and miR-518f-3p obtained by RT-qPCR, these miRs were excluded from further investigation, but the trend observed in the miR profiling for the other selected miRs was confirmed in a single miR analysis. In fact, the miR-122-5p expression level increased while miR92a-3p, miR-30c-5p, and miR-18a-5p decreased after LT.

Then, the analysis of the selected miRs (i.e., miR-122-5p, miR92a-3p, miR-30c-5p, and miR-18a-5p) was extended to the whole cohort of 14 recipients with pre- and post-LT samples, as reported in Table 1. Due to the large range of FU times, the recipients‘ group was divided considering two FU periods (i.e., 4–10 and 11–26 months), and each one included seven different patients paired for pre–post-LT assessment. The rationale of the FU subdivision was based on the dose of the immunosuppressive treatment, since a reduction normally occurs at 10–12 months after LT. This period was considered critical for patients’ health status. Figure 2 shows the results of miR-122-5p and miR-92a-3p, according to the FU times. A significant decrease in late FU (11–26 months) compared to pre-LT was found for both miRs (panel B and panel D, respectively), while no difference was detected between pre-LT and early FU (4–10 months). At this phase of analysis, miR-30c-5p and miR-18a-5p resulted in non-significant modification.

### 2.3. Pre-Post LT and Analysis of Delta Difference

The difference of miR expression between pre- and post-LT at two different FU times (4–10 months vs. 11–26 months) was conducted only for those miRs resulted significantly changed in the validation analysis i.e., miR-122-5p and miR-92a-3p. Figure 3 reports the delta analysis of miR-122-5p (panel A) and miR-92a-3p (panel B). Delta A represents the changes in the miR expression between the pre-LT and early FU, while Delta B represents the changes in the miR expression between the pre-LT and late FU, showing a significant variation according to the FU time.

### 2.4. MiR Analysis after Transplant on the Extended Sample Cohort

The expression of miR-122-5p, miR-92a-3p, miR-18a-5p, and miR-30c-5p was also assessed by RT-qPCR in additional serum samples from 19 LT patients collected only during the FU (as described in Table 2). The data analysis was performed on the whole group of post-LT recipients, including the 14 post-LT samples previously analyzed (Table 1) for a total of 33 specimens. The samples were stratified into three groups for their specific FU time, i.e., 4–10, 11–18, and 19–26 months. This stratification was applied to better monitor the FU time-dependent miR changes, as reported in Figure 4. The MiR expression was also evaluated in 24 serum samples from a control group of age–sex matched, nontransplanted, and healthy subjects (CTR).

MiR-122-5p (Figure 4A) showed an increase in early FU (4–10 months) in comparison with the control group, a decrease in middle FU (11–18 months), and then a further significant increase in late FU (19–26 months). A similar trend but less significant was observed for miR-92a-3p and miR-18a-5p (Figure 4B,C, respectively). On the contrary, miR-30c-5p (Figure 4D) significantly decreased in early and middle FU compared to the control group, while a recovery was found in late FU.

### 2.5. Liver Function Tests (LFTs) along FU Times and Correlation with miR Expression

LFTs, described above, were analyzed at the same FU times adopted for the miR assessment. Neither significant differences were found along the various FUs (Appendix A) nor out of the normal ranges. Correlations were performed between miRs and LFTs in recipients after LT at various FUs. In early FU (4–10 months after LT), miR-122-5p positively correlated with AST, while a negative correlation was found between the following parameters: (1) miR-92a-3p and albumin (ALB); (2) miR-30c-5p and alanine aminotransferase (ALT); (3) miR-30c-5p and gamma-glutamyl transferase (GGT), as reported in Table 3 where significant *p* values are highlighted in bold. In late FU (19–26 months after LT), as displayed in Table 4 (significant *p* values are shown in bold), miR-122-5p positively correlated with GGT and alpha-fetoprotein (ALP), and miR-30c-5p negatively correlated with total bilirubin (TBIL). No correlations were found in middle FU (11–18 months after LT).

### 2.6. Levels of the Selected miRs with or without LT Complications

The patients were also stratified by complication development or no complication after LT, aiming to evaluate the expression level of the selected miRs. In respect to the healthy control subjects, miR-122-5p, miR-92a-3p, and miR-18a-5p significantly increased in the patients with complications, while a significant decrease was detected for miR-30c-5p, as displayed in Figure 5.

Complications after LT were also evaluated based on their severity by assigning a score from one to three, with a maximum value for the greatest severity. No significant miR correlations were raised by means of these analyses, likely due to the limitation in the number of patients for each group.

### 2.7. Bioinformatic Pathway Analysis

The bioinformatic analysis led to the identification of common pathways among miR-122-5p, miR-92a-3p, miR-18a-5p, and miR-30c-5p based on the experimentally validated target. A KEGG pathway analysis was applied considering the molecular pathways, including gene transcripts targeted by all of the considered miRs.

The four pathways resulted in significance, but two out of the four were the most significant (*p* = 2.96 × 10^−8^), i.e., lysine degradation cascade and transforming growth factor-beta (TGF-β) signaling pathways. The former included two gene transcripts as common targets of all of the analyzed miRs, i.e., ASH1-like histone lysine methyltransferase (ASH1L) and lysine methyltransferase 2A (KMT2A). Their action points are present in one specific section of the lysine pathway, as presented in Figure 6A. The latter includes one common mRNA target of all the selected miRs, i.e., SMAD family member 2 (SMAD2). The TGF-β pathway and the sites of action of SMAD2 are reported in Figure 6B.

### 2.8. MiRs of Liver Donor–Recipient Age Mismatches

Following the results of a previous study [11], where miR-31-5p, miR-141-3p, and miR-200c-3p were identified as biomarkers of liver aging, the same miRs were analyzed in the serum samples of all recipients. The results showed that those liver-specific miRs were not detectable in peripheral blood, and high Ct values (Ct > 32) were observed.

A further analysis of the currently selected miRs, considering the age mismatches between donor and recipient, was performed analogously to the previous study, but no significant differences were found.

## 3. Discussion

The current work aimed to investigate circulating miRs before and after LT as potential and sensitive biomarkers for patients’ monitoring. The phase of monitoring is crucial after transplantation, since patients may develop a type of rejection or possible complications, also due to the various immunosuppression therapies and dose adjustments. Thus, the function of the transplanted organ should frequently be monitored, and new circulating and specific biomarkers could be useful, being not technically invasive, as is a liver biopsy.

In this perspective, a discovery phase of miR profiling was performed using arrays on serum samples collected before and after graft. The MiRs selected were validated in the same samples and subsequently in the extended cohort, for a total of 28 sera collected from 14 recipients at pre- and post-LT. Four miRs (i.e., miR-122-5p, miR-92a-3p, miR-18a-5p, and miR-30c-5p) emerged as the most relevant and were also assessed in a different subset of patients collected only after LT (33 specimens) and compared with a control group of 24 age–sex matched, nontransplanted, and healthy subjects. The data were stratified according to the FU period showing significant differences in the miR expression. On the contrary, the data obtained from haemato-biochemical analyses (LFTs) did not show altered levels, or out of the normal physiological range, during the same FU times. These results suggest that LFTs likely lack the specificity and sensitivity to monitor liver function after transplant, while serum miRs could be more sensitive biomarkers to monitor possible outcomes.

As far as rejection is concerned, it occurred in only one patient due to the noncompliance with immunosuppressive therapy. Excluding three patients who did not exhibit any sign, common complications that manifested in more than one patient were the following: infections, primary disease recurrence, kidney failure, biliary complication, incisional hernia, hypertrophic cardiomyopathy, acute coronary syndromes, atrial fibrillation, ascites, drug-induced neurotoxicity, metabolic alterations, and anemia. Others occurring more sporadically were the following: respiratory failure, GvHD, deep vein thrombosis, and embolism. The complications were registered in the first 10 months after LT (early FU) in approximately half of the recipients, while in the middle FU period (11–18 months) only two patients developed problems related to LT. This list of outcomes seems to fit with the trends observed with the identified miRs, in particular for miR-122-5p and miR-92a-3p, considering the increase in their level in the blood in early and late FU as indicative of liver injury/suffering. In this respect, stratifying patients based on the presence or absence of complications after LT (independently from FU times), the recipients with complications presented a higher level of miR-122-5p, miR-92a-3p, and miR-18a-5p and a lower of miR-30c-5p than the physiological level of the same miRs in the control group. The same analyses were performed with the LFTs, but no difference was observed (Appendix A), thus confirming the potential power for complication prediction by adopting the selected miRs and further investigations.

Over the last decade, circulating miRs have become the subject of intense research as promising noninvasive biomarkers in many different pathological conditions, as well as in liver disease [15,16]. In the context of LT, miR signatures in serum or plasma have been studied in order to identify predictive, diagnostic, and prognostic biomarkers [17,18].

The serum miR-122 level was found to be significantly increased concomitantly to rejection signs, and then it decreased 6 months after the resolution of rejection, with a kinetic similar to those of AST and ALT [17]. The serum concentrations of miR-122 were significantly elevated in patients with cholangiocarcinoma compared to healthy controls or patients with primary sclerosing cholangitis without malignant transformation, and a strong postoperative decline in miR-122 serum levels was significantly associated with a favorable patient prognosis [19].

Although no data concerning circulating miR-92a-3p in LT are yet available, Shigoka et al. showed an increased expression of miR-92a-3p in the plasma samples of hepatocellular carcinoma patients after surgical tumor resection, thus suggesting a possible mechanism linking therapy, cellular damage, and stress or miR-specific exocytosis during the first critical months after surgery [20].

As far as miR-18a-5p is concerned, no results related to LT have been obtained until now. However, interestingly, this miR belongs to the same miR cluster of miR-92a-3p, likely explaining the similar trends in both miRs in FU. MiR-18a-5p and miR-92a-3p are members of the miR-17/92 cluster, one of the best characterized and described miR family [21]. This cluster maps to human chromosome 13 and encodes for six individual miRs (i.e., miR-17, miR-18a, miR-19a, miR-20a, miR-19b-1, and miR-92a). Publications have grown exponentially since its discovery, revealing the roles of its members in a wide variety of settings that include normal development, immune diseases, cardiovascular diseases, neurodegenerative diseases, and aging [22,23]. The miR-17-92 cluster was found to be overexpressed in many human cancers and to promote unrestrained cell growth, and it has, therefore, been termed onco-miR-1 [24]. Emerging evidence indicates that the miR-17-92 cluster also plays an important role in liver diseases and carcinogenesis [25]. A liver-specific miR-17-92 transgenic mouse showed a significant increase in hepatocellular cancer development, and an overexpression of the miR-17-92 cluster in cultured human hepatocellular cancer cells enhanced the proliferation, colony formation, and invasiveness in vitro, whereas inhibition of the miR-17-92 cluster had the opposite effect [26]. The overexpression of the miR-17-92 cluster or its members in in vitro cultured human cholangiocarcinoma cells enhanced tumor cell proliferation, colony formation, and invasiveness [27]. Nevertheless, the involvement of this cluster in liver transplant is not yet well defined.

Intriguingly, miR-122-5p, miR-92a-3p, and miR-18a-5p display analogous trends at different FU times, thus suggesting a possible similar dysregulation in terms of a biological role after transplant. The significantly high level of those miRs in early and late FU compared with the control group may occur to some extent due to the fact of liver dysfunction/slow chronic rejection and eventually long-term immunosuppression-induced toxicity. This last hypothesis is supported by different evidence on circulating miR-122-5p used as biomarker of drug-induced toxicities [28].

Differently, serum miR-30c-5p expression is significantly lower in recipients after LT in early and middle FU, while at advanced FU the expression level becomes similar to healthy controls, as a sort of recovery from the downregulated status. These results may suggest a possible implication of miR-30c-5p in immunosuppression/tolerance induction to transplant. Several studies show the role of miR-30c-5p in liver diseases, such as in HCV-positive cirrhosis [29], in hepatic steatosis [30], and in hepatocellular carcinoma not only for the improvement of diagnosis but also for a prognostic and therapeutic approach [31,32].

The correlations between circulating miRs and LFTs confirm the importance of these biomarkers in early and late FU. Positive correlations with miR-122-5p were found with AST in early FU and with GGT and ALP in late FU. These findings are in accordance with previously published data, showing positive correlations between circulating miR-122-5p levels and serum transaminases and GGT [17,33,34], even if some evidence suggests that miR-122-5p is more specific and sensitive than the standard ALT and AST enzymes [35].

On the contrary, miR-30c-5p negatively correlates with ALT and GGT in early FU, and with TBIL in late FU, thus marking a different trend and, likely, a diverse role of this miR in liver function after transplant.

A bioinformatic analysis was applied to identify common pathways among significant miRs, i.e., miR-122-5p, miR-92a-3p, miR-18a-5p, and miR-30c-5p. Computational KEGG pathway analysis identified lysine degradation and TGF-β signaling as the most significant pathways. In particular, miRs shared target genes/mRNAs crucial for chromatin remodeling (i.e., ASH1-like histone-lysine methyltransferase (ASH1L) and histone-lysine N-methyltransferase 2A (KMT2A)), thus suggesting a relevant role linked to the lysine-mediated epigenetic changes. As a matter of fact, posttranslational modifications of lysine have a crucial importance in histones epigenetic regulation, in both physiological and pathological processes, such as in liver diseases [36]. In accordance, a study highlighted histone lysine methylation as a fundamental epigenetic mechanism in liver regeneration [37]. Therefore, variations in miR levels may result in the modification of a great number of histone-related epigenetic mechanisms, with consequences on a large spectrum of processes in liver pathophysiology. Other epigenetic modifications in terms of the DNA methylation profile were investigated, and a large remodeling of DNA methylation patterns have been found [38]. Liver-specific age-related changes, such as histone modifications, DNA methylation, miRs, N-glycan profiles, serum metabolites, gut microbiome species, and their products, were proposed to better identify the biological ages of both liver donors (at organ level) and recipients (at systemic level) in an LT context [39].

Interestingly, the current work suggests SMAD family member 2 (SMAD2) as common target of the selected miRs, known to be key regulator of TGF-β signaling pathway. The involvement of this pathway in liver disease has already been described, and it is known to contribute to different and essential cellular processes for homeostasis, such as proliferation, differentiation, migration, and cell death [40]. TGF-β is the most well-known hepatocyte proliferation inhibitor and stops the signal in the process of hepatic regeneration, whereas SMAD2 is one of the regulators of this function [41]. TGF-β has been shown to play an essential role in establishing immunological tolerance in transplantation [42], suggesting the potential effects of the selected miRs for graft tolerance and transplant outcomes.

Lastly, specific miRs emerged as markers of liver aging from a previous work [11], i.e., miR-31-5p, miR-141-3p, and miR-200c-3p. They were also investigated in the serum of the same patients in the current work, but the lack of a consistent expression of those miRs in the blood have highlighted their role as organ-specific markers only. Furthermore, the selected miR-122-5p, miR-92a-3p, miR-18a-5p, and miR-30c-5p were also analyzed considering the age-mismatches between donor and recipient, but no difference was found, thus suggesting that the chronological age of the liver donors did not affect the serum level of those miRs.

Some limitations of the work are the following: (1) the relative short period of FU; (2) the unique time point for most recipients; (3) the onset of complications not matching with the FU times that were planned by surgeons and standard protocol. These limitations are due to the complex setup of a human transplant context, where patients’ health is the primary objective definitively.

In conclusion, the current work proposes miR-122-5p, miR-92a-3p, and miR-18a-5p as potential biomarkers (two out of three belong to the same miR-17-92 cluster), being more sensitive than canonical liver function parameters, for monitoring patients after LT and possible hepatic injury. The recovery of miR-30c-5p along the FU times does not suggest the use of this miR as potential biomarker.

## 4. Materials and Methods

### 4.1. Experimental Design and Study Participants

The enrolment of patients was conducted following the protocol of donor–recipient allografts performed at the Hepatobiliary and Transplant Surgery Unit (IRCCS, S. Orsola Hospital, Bologna, Italy) with the ethical committee’s approval (code: 44/2008/Tess) in the framework of a national project (PRIN2008), several years before the SARS-CoV-2 pandemic period. Serum samples were collected from 14 LT recipients before surgery (pre-LT) and during the follow-up period (FU) at different times after transplantation, for a total of 28 samples. The main characteristics of this group of patients are reported in Table 1.

The second group of samples was collected from recipients only after LT in the FU period (N = 19). The main characteristics of this cohort are summarized in Table 2. The same serum samples, for both cohorts, were analyzed and described in detail in the previous publication [11], even if some serum samples from those patients are currently no longer available. Thus, the total number of enrolled recipients was 33, their age range was from 25 up to 69 years, and their average age was of 50.6 ± 10.5 (years ± SD).

Furthermore, serum samples collected from a nontransplanted control group (N = 24) were evaluated as a reference for all of the investigated miRs. The control group was constituted by 28–68 year old healthy subjects, average 49.3 ± 12.3 (years ± SD), matching the age and sex of the recipients.

Liver function markers were acquired for all patients enrolled in the study, in collaboration with the General Surgery and Transplant Unit (S. Orsola Hospital, Bologna, Italy). In particular, total bilirubin (TBIL), indirect bilirubin (IBIL), aspartate aminotransferase (AST), alanine aminotransferase (ALT), gamma-glutamyl transferase (GGT), alpha-fetoprotein (ALP), and albumin (ALB) were acquired. These data were also described in a previous study [11].

### 4.2. RNA Extraction

Total RNA was isolated from 100 μL of serum using the Total RNA purification kit (Norgen Biotek Corporation, Thorold, ON, Canada). The protocol was modified adding 20 fmol of cel-miR-39 (Qiagen, Hilden, Germany) at the lysis step as a spike-in control to verify the RNA isolation efficiency.

### 4.3. MicroRNA Profiling

Human miR microfluidic card, TaqMan Array Human MicroRNA A Card (Applied Biosystems by Thermo Fisher Scientific, Waltham, MA, USA), enabling the quantitation of 377 human miRs, was used to identify miR-profiles in 6 serum samples from 3 recipients before and after LT (the first three patients in Table 1 are highlighted in bold).

RNA was converted to cDNA by priming with a mixture of looped primers and then pre-amplified using the MegaPlex primer pools (Applied Biosystems by Thermo Fisher Scientific, Waltham, MA, USA), according to the manufacturer’s instructions. The profiling was performed using an Applied Biosystems 7900 HT real-time PCR instrument.

The MiR profiling was normalized using the median of the overall miR expression on each array (ΔCt). Only the miRs expressed in all of the samples were selected for analyses, and Ct values ≤ 30 were established as the cut-off. The fold change (2^−ΔΔCT^) was calculated based on the estimated mean difference between vascular groups. Fold changes ≥ 2 and ≤−2 were selected.

### 4.4. Validation in RT-qPCR

The RT-qPCR was performed through TaqMan technologies (Thermo fisher scientific, Waltham, MA, USA) following the manufacturer’s protocol. The RNA was transcribed to cDNA with the TaqMan MicroRNA Reverse Transcription Kit (Applied Biosystems, Thermo Fisher Scientific, Waltham, MA, USA), and the real-time quantitative PCR (RT-qPCR) was subsequently performed with TaqMan MicroRNA Assays. The data were normalized to the cel-mir-39 measured in each sample, and the relative expression was calculated with the delta Ct method.

### 4.5. Statistical Analysis

The statistical analyses for the miR data were performed with a nonparametric test by means of IBM SPSS software, version 26 (IBM Corp., Armonk, NY, USA), and a *p*-value ≤ 0.05 was considered statistically significant.

A wilcoxon nonparametric test was conducted on paired values to compare the miR expression before and after transplant. A delta analysis was performed on the pre-LT and FU paired samples, depending on the FU timing, and Mann–Whitney nonparametric tests were performed. The Kruskal–Wallis nonparametric test was applied among the groups, considering the miR expressions and liver function tests (LFTs).

In addition, correlations between validated miR expression and LTFs (i.e., TBIL, IBIL, AST, ALT, ALP, GGT, and ALB) were performed using Spearman’s rank-order correlation tests.

### 4.6. Bioinformatic Pathway Analysis

DIANA-miRPath v3.0 (http://www.microrna.gr/miRPathv3, accessed on 1 November 2021) was used to identify the validated targets of the miRs and to find common pathways among them as well. Only experimentally validated interactions were considered, and KEGG pathway analysis was applied using the “genes intersections” mode, thus discriminating molecular pathways that included gene transcripts targeted by all of the analyzed miRs.

## Figures and Tables

**Figure 1 ijms-24-03457-f001:**
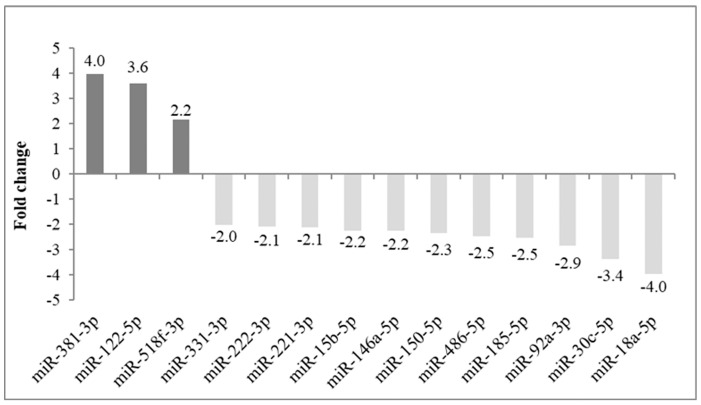
Pre- vs. post-LT serum miR profiling: up- and downregulated miRs. MiRs showing relevant (fold change ≥ 2) upregulation are displayed in dark grey (N = 3), while miRs with relevant downregulation (fold change ≤ −2) are displayed in light grey (N = 11) in three recipients analyzed before and after transplantation. Fold change values are reported for each miR.

**Figure 2 ijms-24-03457-f002:**
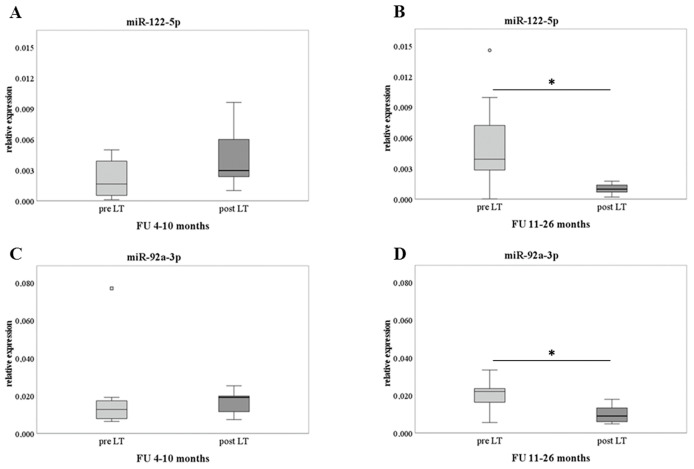
Validation analysis on pre- and post-LT samples. RT-qPCR results of miR-122-5p (panel (**A**,**B**)) and miR-92a-3p (panel (**C**,**D**)) are shown as box plots (with median) of the relative expression for each group. The analyses were performed comparing different FU periods (4–10 and 11–26 months) and assessing 7 patients for each group. Paired samples data were analyzed with Wilcoxon Mann-Whitney tests. * *p* ≤ 0.05.

**Figure 3 ijms-24-03457-f003:**
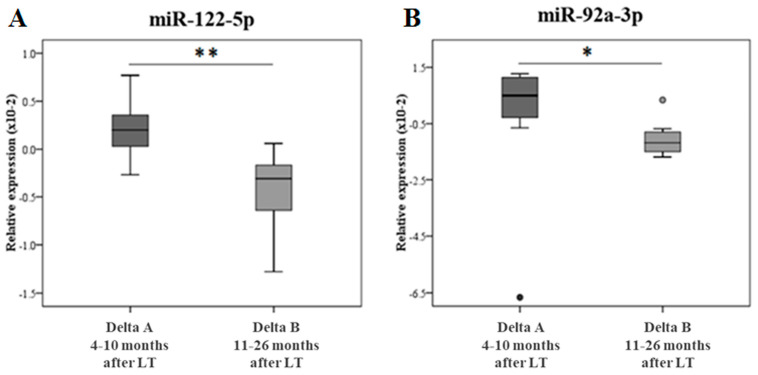
Delta (pre- vs. pos-LT) analyses of the miR relative expression at different FU periods. In panel (**A**), the miR-122-5p delta of the relative expression; in panel (**B**), the miR-92a-3p delta of the relative expression at different FUs are shown. Delta A represents the changes in the miR relative expression between pre- and 4–10 months post-LT in the same subjects (N = 7). Delta B represents the changes in the miR relative expression between pre-LT and 11–26 months post-LT in the same subjects (N = 7). The data are reported as box plots (with median) and were analyzed using the Mann–Whitney nonparametric test. * *p* ≤ 0.05 and ** *p* ≤ 0.01.

**Figure 4 ijms-24-03457-f004:**
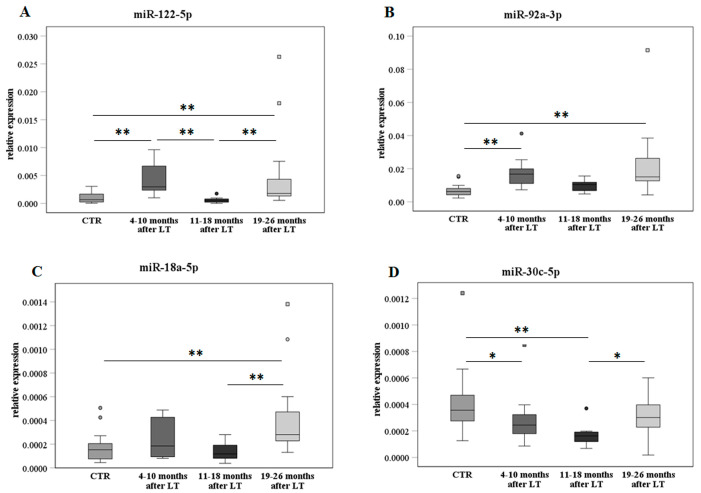
MiR expression analyses according to the FU and compared to the nontransplanted healthy control group. MiR-122-5p (panel (**A**)), miR-92a-3p (panel (**B**)), miR-18a-5p (panel (**C**)), and miR-30c-5p (panel (**D**)) are shown as box plots (with median) of the relative expression for each group along FU, i.e., 4–10 (N = 11), 11–18 (N = 8), and 19–26 months (N = 14). The data were also compared to the nontransplanted healthy control group (CTR, N = 24) and analyzed with the Kruskal–Wallis test. * *p* ≤ 0.05 and ** *p* ≤ 0.01.

**Figure 5 ijms-24-03457-f005:**
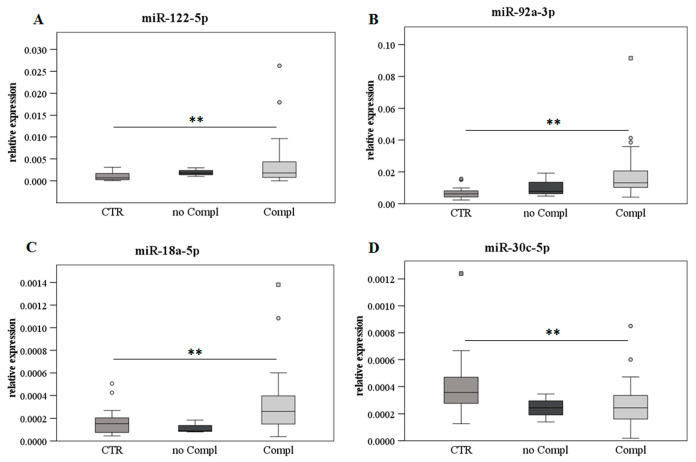
Levels of the selected miRs with or without LT complications. The data on the miR expression level in the presence (N = 30) or absence (N = 3) of complications after LT in comparison with the healthy control group (CTR, N = 24) are reported. MiR-122-5p (panel (**A**)), miR-92a-3p (panel (**B**)), miR-18a-5p (panel (**C**)), and miR-30c-5p (panel (**D**)) are shown as box plots (with median) of the relative expression for each group. The data were analyzed with the Kruskal–Wallis test. ** *p* ≤ 0.01.

**Figure 6 ijms-24-03457-f006:**
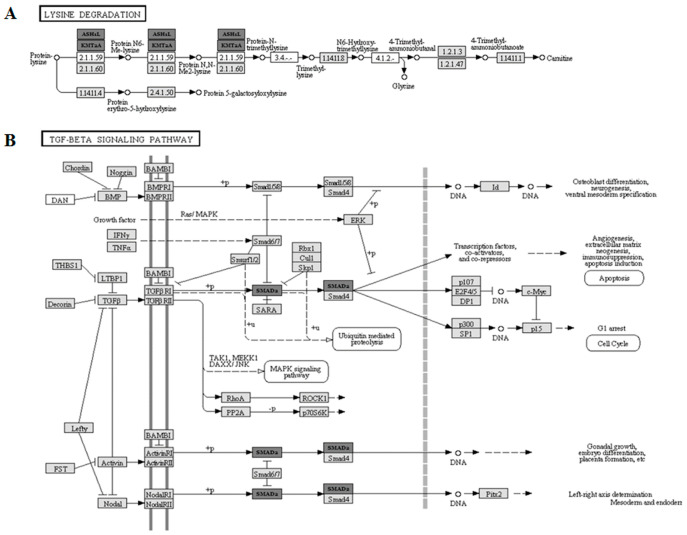
Bioinformatic pathway analyses. Only experimentally validated interactions were considered by means of KEGG. The bioinformatic analyses identified 2 out of 4 pathways as being the most significant (panel (**A**,**B**); *p* = 2.96 × 10^−8^) including gene transcripts targeted by all the selected miRs (miR-122-5p, miR-92a-3p, miR-18a-5p and miR-30c-5p). (Panel (**A**)) A section of the lysine degradation pathway is reported, where the targets ASH1-like histone lysine methyltransferase (ASH1L) and lysine methyltransferase 2A (KMT2A) are described at their points of action. Their key role in chromatin-remodeling/lysine methylations is outlined. (Panel (**B**)) TGF-β signaling is displayed with the common target of all selected miRs (i.e., SMAD family member 2 (SMAD2)) outlining its essential role as an inhibitory pathway.

**Table 1 ijms-24-03457-t001:** Characteristics of the donors and recipients. The recipients’ serum samples were collected before (pre-LT) and after liver transplant (FU).

Sample Code	Donor Age	Recipient Age	Donor Sex	Recipient Sex	Recipient Disease	FU (Months)
**78**	**71**	**40**	**M**	**M**	**HCV-cirrhosis**	**8**
**84**	**49**	**49**	**M**	**M**	**HCC with HBV-cirrhosis**	**7**
**88**	**12**	**45**	**M**	**M**	**HCC with HCV-cirrhosis**	**4**
51	70	26	M	M	HBV and HDV-cirrhosis	7
54	58	46	F	M	HBV and HDV-cirrhosis	12
55	68	50	M	M	HCV-cirrhosis	5
56	74	50	F	F	Alcoholic cirrhosis	13
81	89	55	M	M	HCC with HBV and HDV-cirrhosis	18
86	53	69	M	M	HCC with HCV-cirrhosis	6
91	87	34	M	M	Budd–Chiari syndrome	26
96	83	63	M	M	HCC with HCV-cirrhosis	23
99	73	48	M	F	HCC with HCV-cirrhosis	7
114	83	46	F	M	HBV and HDV-cirrhosis	20
121	12	54	F	M	Primary biliary cirrhosis	14

HBV, hepatitis B virus; HCC, hepatocellular carcinoma; HCV, hepatitis C virus; HDV, hepatitis D virus.

**Table 2 ijms-24-03457-t002:** Characteristics of the recipients enrolled after transplant (FU serum samples only).

Sample Code	Donor Age	Recipient Age	Donor Sex	Recipient Sex	Recipient Disease	FU (Months)
1	59	62	M	M	HCC with HCV-cirrhosis	20
3	69	51	F	M	Cryptogenic cirrhosis	8
6	43	57	M	M	Alcoholic cirrhosis	23
9	76	42	F	F	Alcoholic cirrhosis	19
10	37	52	M	M	HCV-cirrhosis	19
17	23	46	M	M	HCC with HCV and alcoholic cirrhosis	22
19	87	60	M	M	Alcoholic cirrhosis with HCC	20
22A	29	36	M	M	HCC with HBV and HDV-cirrhosis	17
22B	29	56	M	M	Polycystic liver and kidney disease	17
24	50	66	M	M	HCC with HCV-cirrhosis	18
39	20	49	M	M	Amyloidosis	8
40	74	66	M	F	HCC with HCV-cirrhosis	11
41	69	64	F	M	HCC with HCV-cirrhosis	22
43	74	56	F	M	Alcoholic cirrhosis	22
45	45	45	M	M	HCV-cirrhosis	26
46	75	57	M	M	Alcoholic cirrhosis with HCC	10
48	78	25	F	M	Primary sclerosing cholangitis	8
104	88	54	F	F	HBV and HDV-cirrhosis	25
107	64	52	M	F	Polycystic liver and kidney disease	23

HBV, hepatitis B virus; HCC, hepatocellular carcinoma; HCV, hepatitis C virus; HDV, hepatitis D virus.

**Table 3 ijms-24-03457-t003:** Correlation analysis between miRs and LFTs at early FU (4–10 months).

Sperman’s Rho		miR-122-5p	miR-92a-3p	miR-18a-5p	miR-30c-5p
TBIL	Correlation coefficient	0.52	−0.10	0.26	−0.19
	Sig. (2-tailed)	0.10	0.77	0.44	0.58
	N	11	11	11	11
IBIL	Correlation coefficient	0.49	0.10	0.44	−0.14
	Sig. (2-tailed)	0.13	0.76	0.17	0.67
	N	11	11	11	11
AST	Correlation coefficient	0.71	−0.44	−0.19	−0.54
	Sig. (2-tailed)	**0.01**	0.17	0.58	0.09
	N	11	11	11	11
ALT	Correlation coefficient	0.50	−0.20	−0.12	−0.72
	Sig. (2-tailed)	0.12	0.55	0.73	**0.01**
	N	11	11	11	11
GGT	Correlation coefficient	0.43	−0.35	−0.35	−0.71
	Sig. (2-tailed)	0.22	0.33	0.32	**0.02**
	N	10	10	10	10
ALP	Correlation coefficient	0.49	0.21	−0.20	−0.33
	Sig. (2-tailed)	0.15	0.56	0.59	0.35
	N	10	10	10	10
ALB	Correlation coefficient	−0.35	−0.91	0.55	−0.07
	Sig. (2-tailed)	0.45	**0.00**	0.20	0.88
	N	7	7	7	7

TBIL, total bilirubin; IBIL, indirect bilirubin; AST, aspartate aminotransferase; ALT, alanine aminotransferase; GGT, gamma-glutamyl transpeptidase; ALP, alkaline phosphatase; ALB, albumin.

**Table 4 ijms-24-03457-t004:** Correlation analysis between miRs and LFTs at late FU (19–26 months).

Sperman’s Rho		miR-122-5p	miR-92a-3p	miR-18a-5p	miR-30c-5p
TBIL	Correlation coefficient	0.15	−0.20	−0.12	−0.73
	Sig. (2-tailed)	0.62	0.51	0.69	**0.00**
	N	13	13	13	13
IBIL	Correlation coefficient	0.07	−0.32	0.01	−0.53
	Sig. (2-tailed)	0.82	0.29	0.96	0.06
	N	13	13	13	13
AST	Correlation coefficient	0.45	0.09	−0.33	0.15
	Sig. (2-tailed)	0.13	0.77	0.27	0.63
	N	13	13	13	13
ALT	Correlation coefficient	0.53	−0.03	−0.11	0.34
	Sig. (2-tailed)	0.06	0.92	0.72	0.26
	N	13	13	13	13
GGT	Correlation coefficient	0.57	−0.40	−0.04	0.10
	Sig. (2-tailed)	**0.04**	0.18	0.90	0.75
	N	13	13	13	13
ALP	Correlation coefficient	0.65	−0.24	−0.05	−0.37
	Sig. (2-tailed)	**0.02**	0.42	0.88	0.22
	N	13	13	13	13
ALB	Correlation coefficient	0.29	−0.23	0.46	−0.44
	Sig. (2-tailed)	0.58	0.66	0.36	0.38
	N	6	6	6	6

TBIL, total bilirubin; IBIL, indirect bilirubin; AST, aspartate aminotransferase; ALT, alanine aminotransferase; GGT, gamma-glutamyl transpeptidase; ALP, alkaline phosphatase; ALB, albumin.

## Data Availability

Data are reported in detail in the current paper. Other data obtained in the same project can be found in Capri et al., 2017 [11]; Bacalini et al., 2019 [38] (see references).

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
