# Peer review of "Circulating miR-122-5p, miR-92a-3p, and miR-18a-5p as Potential Biomarkers in Human Liver Transplantation Follow-Up"

_ijms, 2023, doi:10.3390/ijms24043457_

Round 1
Reviewer 1 Report
In themanuscript entitled "Circulating miR-122-5p, miR-92a-3p and miR-18a-5p as potential biomarkers in human liver transplantation follow-up ", the authors profiles the miR of patients before and after LT. They detected some miRs with significant change during the whole follow up peroid. while such information is potentially useful, the manuscript didnot clearly state what is the application of this observation. Like in the title "as potential biomarkers", it is not clear a biomarker of what.
The data within the paper is observational, this is fine, but the authors need to do deeper analysis to address the meaning of these changes. For example, can the markers used to predict prognostic or adverse events, etc, such predition need to be scientifically analyzed.
minor
In the result part, please mention the type of analysis performed, not just saying "the analysis".
Figure 1 should have error bars.
Author Response
Reviewer 1
Comments and Suggestions for Authors
In the manuscript entitled "Circulating miR-122-5p, miR-92a-3p and miR-18a-5p as potential biomarkers in human liver transplantation follow-up ", the authors profiles the miR of patients before and after LT. They detected some miRs with significant change during the whole follow up peroid. while such information is potentially useful, the manuscript didnot clearly state what is the application of this observation. Like in the title "as potential biomarkers", it is not clear a biomarker of what.
-The data within the paper is observational, this is fine, but the authors need to do deeper analysis to address the meaning of these changes. For example, can the markers used to predict prognostic or adverse events, etc, such prediction needs to be scientifically analyzed.
Authors would like to thank the reviewer for this important observation. The current work proposes the identified miRs i.e., miR-122-5p, miR-92a-3p and miR-18a-5p, as evaluable biomarkers of transplant monitoring since all these miRs increased in all patients who developed complications (described at page 8), while standards hematobiochemical analysis did not (supplementary figure). The characteristic trends of the above mentioned miRs along various follow-up time, overlap the trends of important liver parameters, such as total and indirect bilirubin, but only miRs values result significant. These findings suggest the major sensitivity of miRs if compared with standard liver function parameters. However, the possibility of prediction of adverse event, a very important task, cannot be apply at the moment since the miRs changes were not detected simultaneously to the complication onset, as described in the limitation of the work (page 11). This crucial issue will be a part of a new project with additional funds.
The literature indicates the potential role of the proposed miRs as biomarkers of liver injury after transplantation or in other hepatic pathologies as well as described in the discussion. In this respect, our findings reinforce this role for the identified miRs.
Minor
-In the result part, please mention the type of analysis performed, not just saying "the analysis".
According to referee’s suggestion, authors have now specified which type of analysis and have improved the result section.
-Figure 1 should have error bars.
Authors thank the reviewer for pointed this out. Nevertheless, based on authors experience and standard analysis of miR profiling by means of card array, the bar could not show the SD/errors, since this first analysis wants to raise up relevant miRs to be furthermore validated in single RT-qPCR. In addition, the fold change is calculated on the means of delta values of each group (in this case 3 recipients before liver transplantation and the same recipients after transplantation). Please see also our previous work/paper showing this well accepted procedure for miR profiling only (Kangas R, Morsiani C, Pizza G, Lanzarini C, Aukee P, Kaprio J, Sipilä S, Franceschi C, Kovanen V, Laakkonen EK, Capri M. Menopause and adipose tissue: miR-19a-3p is sensitive to hormonal replacement. Oncotarget. 2017 Dec 18;9(2):2279-2294. doi: 10.18632/oncotarget.23406).

Reviewer 2 Report
In the current work Authors propose miR-122-5p, miR-92a-3p and miR-18a-5p as potential biomarkers for monitoring patients after LT and liver ischemia reperfusion damage. Interestingly They also demonstrate that others MiR, such as miR-30c-5p, does not a potential biomarker. Although the present study is potentially easily understandable and very interesting, there are several issues that should be addressed as noted below.
_There are several limitations in this work that may be improved by increasing the number of patients included in this study the most important are 1) the unique time point for most recipients and 2) the relative short period of follow-up times. I understand also that there are other limitations but most of them are due to the complex set up of LT context.
- miRs as invasive approach to evaluate liver damage and they have gained enormous interest in the field of biomarker of liver ischemia reperfusion and LT authors should cited others relevant papers in the introduction section, here only three papers are cited in this part.
- In the characteristics of donors there are marginal livers included in this study? used of mild and moderate steatosis (most important source of organ donors) may enriched the paper and may generate other data.
_In discussion section based on the results I believe that Authors propose some possible modulation of pathways to reduce Liver damage especially when machine perfusion are used to preserve graft.
Minor concern: The manuscript would benefit some from additional English-language editing.
Author Response
Reviewer 2
Comments and Suggestions for Authors
In the current work Authors propose miR-122-5p, miR-92a-3p and miR-18a-5p as potential biomarkers for monitoring patients after LT and liver ischemia reperfusion damage. Interestingly They also demonstrate that others MiR, such as miR-30c-5p, does not a potential biomarker. Although the present study is potentially easily understandable and very interesting, there are several issues that should be addressed as noted below.
-There are several limitations in this work that may be improved by increasing the number of patients included in this study the most important are 1) the unique time point for most recipients and 2) the relative short period of follow-up times. I understand also that there are other limitations but most of them are due to the complex set up of LT context.
Authors understand the referee’s point of view. However, the complex context of human liver transplantation and the acknowledged closed grant (Italian National Project- PRIN08), do not allow the authors to implement the number of recipients, underlying that all the serum samples available have been completely analyzed. All the limitations, including those mentioned by the referee, have been declared in the text (page 11).
- miRs as invasive approach to evaluate liver damage and they have gained enormous interest in the field of biomarker of liver ischemia reperfusion and LT authors should cited others relevant papers in the introduction section, here only three papers are cited in this part.
Authors thank the referee for the comment and suggestion. Authors agree on the role of miRs in hepatic ischemia-reperfusion injury (HIRI) and possible connection with blood circulating miRs. However, none of the patients developed HIRI, as described in the discussion session (page 8).
As standard protocol at the Unit of Transplant-S.Orsola Policlinic in Bologna (Italy), livers in heart beating and brain-death donors, are monitored before explant for vascular system by means of eco-doppler and after implant, the patient is monitored by the production of bile (applying a specific tube).
Even if, the biological question focused on HIRI does not fully match with the objective of the current work we have now added a reference of the review on HIRI and hepatic miRs, in the introduction “Zhu SF, Yuan W, Du YL, Wang BL. Research progress of lncRNA and miRNA in hepatic ischemia-reperfusion injury. Hepatobiliary Pancreat Dis Int. 2022 Jul 30:S1499-3872(22)00184-9. doi: 10.1016/j.hbpd.2022.07.008.”
- In the characteristics of donors there are marginal livers included in this study? used of mild and moderate steatosis (most important source of organ donors) may enriched the paper and may generate other data.
Authors thank the referee to give the opportunity to better explain the marginal donors’ definition. In fact, all old donors (age > 70 yrs) are considered marginal. The livers obtained from old donors usually have some histological alterations (not only steatosis), that were investigated in a previous work, i.e. Bellavista E, Martucci M, Vasuri F, Santoro A, Mishto M, Kloss A, Capizzi E, Degiovanni A, Lanzarini C, Remondini D, Dazzi A, Pellegrini S, Cescon M, Capri M, Salvioli S, D'Errico-Grigioni A, Dahlmann B, Grazi GL, Franceschi C. Lifelong maintenance of composition, function and cellular/subcellular distribution of proteasomes in human liver. Mech Ageing Dev. 2014 Nov-Dec;141-142:26-34. doi:10.1016/j.mad.2014.09.003)
Histological alterations are always tested before implant by biopsies analysis (also described in Capri et al., 2017; Aging Cell, in the reference list). Livers with confluent necrosis and/or macro vesicular steatosis > 60% and/or fibrosis with f3 level (or level f4 following METAVIR classification) are discarded as standard procedure. The above-mentioned articles have been focused on the same livers, that were implanted in the recipients who are the objective of the current work. In particular, the histological results as published in Bellavista et al., 2014-MAD, are described as follows:
“The histological analysis of the 54 liver biopsies (2 out of 56 were not available) showed the occurrence of a mild portal chronic infiltrate in 17 (31%) cases, with no piecemeal necrosis or lobular necrosis. Fibrosis was scored as absent in 14 (26%) cases, Ishak grade 1 in 19 (35%) and Ishak grade 2 in 21 (39%), myointimal thickening was present in 33 (61%) cases, biliocyte regression in 15 (37%), signs of mild cholestasis in 3 (5%), sinusoid dilatation in 8 (15%) cases. Hepatocytic polymorphism was absent in 18 (33%) cases, mild in 26 (48%) and pronounced in 10 (18%) cases. A moderate-to-severe lipofuscin accumulation was observed in 20 (37%) cases. Some amount of steatosis was present in 29 cases, with a mean percentage of 5.42 ± 10.7 (range 0–60%)”.
Actually, the stratification of all possible liver parameters, including steatosis, decreases the power of analysis and strongly impact on significant results/association with selected miRs and, importantly, all the hematobiochemical analyses resulted in the normal range, as reported at page 6 and supplementary material).
-In discussion section based on the results I believe that Authors propose some possible modulation of pathways to reduce Liver damage especially when machine perfusion are used to preserve graft.
Authors do not focus their work on the effect of liver reperfusion damage, but first objective was to identify blood circulating miRs for patients’ monitoring independently from specific outcomes. Bioinformatic analysis revealed two molecular pathways that suggest the regulation of specific targets that may have a systemic effect. At the current stage, it is difficult to think to possible therapeutic targets.
Minor concern
-The manuscript would benefit some from additional English-language editing.
Authors confirm to have entirely reviewed the manuscript for English language, text and style improvements.
Round 2
Reviewer 1 Report
I think the authors addressed my concerns,recommend accept。